# Research on the Influencing Factors of Cultural and Tourism Service Quality in Huizhou Area

**Xin Wang** [1,2], **Zimeng Yang** [1,2] **and Yanlong Guo** [3,*]

[1] Anhui Cultural Tourism Innovative Development Research Institute, Anhui Jianzhu University, Hefei 230601, China; wangxin020015@ahjzu.edu.cn (X.W.); yzimeng357@gmail.com (Z.Y.)
[2] School of Arts, Anhui Jianzhu University, Hefei 230601, China
[3] Social Innovation Design Research Centre, Anhui University, Hefei 203106, China
[*] Correspondence: 20106@ahu.edu.cn; Tel.: +86-152-5655-6306

**Abstract:** This study investigates the elements that influence the quality of cultural and tourism services in the Huizhou region, seeking to improve the region's tourism competitiveness and revive the growth of the local rural tourism industry based on the background of cultural and tourism integration. This article builds an evaluation index system for the influencing variables of culture and tourism service quality by choosing 20 indicators from three categories: public service quality, cultural service quality, and tourism industry performance. The entropy weight Topsis method is employed to assess the service quality of three districts and five counties in the Huizhou area, which serves as the research object. Research has shown that (1) the number of five-star hotels, the number of A-level scenic spots, and the number of overseas tourists received have a significant weight, and these have become important factors affecting the effective supply of high-quality rural tourism; (2) there is a significant difference in the quality of cultural and tourism services in different areas across the Huizhou region, with Tunxi District and Yi County having the highest level of service quality, while Qimen County and Jixi County have the lowest level; (3) The overall quality of cultural and tourism services in Huizhou region is relatively high, with six districts and counties rated medium or above, accounting for about 75% of the total. Therefore, it is necessary to improve the infrastructure construction, improve the quality of public services, maintain the local and regional culture, accelerate the integration of culture and tourism, and enhance the high-quality development of regional tourism in Huizhou.

**Keywords:** Huizhou; rural cultural tourism; service quality; entropy weight TOPSIS

## 1. Introduction

Huizhou is an important part of Chinese civilization. It is a traditional, ancient Chinese village with a profound cultural heritage, where an excellent farming culture has formed in the daily life of the ancient Huizhou people. Based on the unique local folk culture of Huizhou, it has become one of the popular attractions for rural tourism in China, with Xidi, Hongcun, Chengkan, and other ancient villages in Huizhou also becoming the center of such tourism [1]. Traditional villages constitute a rural cultural tourism system with rich cultural heritage resources and tourism development potential, but traditional villages also have problems, such as differences in resource advantages caused by differences in geographical distribution. In addition to its resource endowment, the development of rural cultural tourism is also affected by the quality of local services [2].

In the last decades, tourism has become extremely important economically as it has become one of the fastest-growing sectors in the contemporary business environment. In 1950, the number of international tourists was 25 million, which increased to about 278 million by 1980, to 682 million by 2000, and to 1.46 billion by 2019 [3,4]. Due to the

maturity of the current tourism market and changes in tourists' preferences, rural tourism is growing rapidly [5] and integrates local culture and heritage maintained by village people to revitalize traditional villages affected by urbanization and modernization [6]. Rural tourism is characterized by the following aspects: safe location, sustainability, community-based features and experiences [7]. Rural tourism is a tertiary sector of economic development [6], and the countryside retains a more pristine natural environment that attracts city dwellers tired of fast-paced life and modern pollution [8]. In many countries, rural tourism has been recognized as a vehicle and an important tool for the development of rural areas [9,10]. In Europe, rural tourism has long been recognized as an effective means of overcoming the social and economic challenges faced by rural areas, which have been associated with the depression of the traditional agricultural industry for more than a century [9,11]. Rural areas are seen as the antithesis of urban areas [12], which is precisely why they are suitable for health protection: they offer opportunities for isolation, relaxation, outdoor activities, and proximity to nature [13–15]. In China, the world's most populous agrarian society, rural tourism has also been widely encouraged in recent decades to revitalize the rural economy and promote rural renewal [9,16]. However, many traditional villages in China have missed the opportunity for development and economic transformation due to insufficient management and service awareness [6]. Therefore, it is necessary to dig deeper into the factors that influence service quality in tourism.

Tourism is essentially a service industry, or more accurately, a mixture of service industries. Tourism service quality, which has been studied since the 1980s, is defined as the degree of satisfying the needs of expressed and potential customers, which directly affects the rise and fall of the local tourism industry. In the context of the tourism industry, tourism service quality is defined as the perception of tourists of services and their quality and the extent to which tour operators and tourism management provide services that meet the needs of tourists [17,18]. Service quality in tourism consists of 10 dimensions, namely core tourism experience, information, hospitality, fair price, amenities, value for money, health, logistics, food, and safety [19]. Tourism service quality is a four-dimensional (4D) structure, and the 4D model consists of four dimensions: assured responsiveness, physical amenities-empathy, reliability, and dependability-directional quality. The four dimensions of tourism service quality have a positive and significant impact on destination image [20], thus having a direct correlation with local tourism. Thus, the study of the factors influencing the quality of cultural and tourism services in Huizhou has important practical significance for the protection of Huizhou's heritage, the improvement of human habitat, and the revitalization of traditional villages [21,22].

Huizhou has "one government and six counties" as the core cultural tourism area [6], but there are service quality problems such as outdated infrastructure, inconvenient transportation, and security problems [23], which directly affect the development of tourism [24,25]. According to Chinese scholars, in carrying out tourism activities, the endowment of the village is the fundamental force [26], the market factor is the decisive force, the government plays a leading role, and other factors, such as transportation and festivals, are facilitating forces [27]. The quality of rural tourism public services and the image of the destination have an impact on the loyalty of tourists [28]. In terms of upgrading, quality, and efficiency based on tourists' tourism experience [27,29], it is also an inevitable requirement to promote the happiness of tourists [30]. Foreign scholars began to study the impact of service quality on the tourism industry earlier; Ilhami Tuncer put forward the customer-perceived service quality model [31], the introduction of service quality theory in the study of tourism, the definition of the concept of tourism service quality from the perspective of tourists. Foreign scholars evaluated the quality of tourism services in Silesian museums and published in academic journals utilizing the SERVQUAL model [32]. Subsequently, Regina Scheyvens suggested that the real needs of tourists are difficult to understand by service providers and that attention should be paid to the tourists themselves in order to improve the quality of tourism services [33]. In the

case of tourism services, quality management is still seen as a marginal activity, and although the importance of the issue is understood, there is a lack of knowledge in terms of concrete actions that can be taken [34]. On the other hand, it is quality that shapes the tourism brand [35], and to continuously improve the quality of the services provided, it is necessary to assess them on a regular basis [36]. Identifying appropriate assessment techniques in certain industries can improve the perception of service quality and contribute to its development, subsequently applying these techniques to other industries [37]. In addition to quantitative research, qualitative research also plays a key role, with both tourist satisfaction and IWOM mediating the relationship between service quality and destination loyalty [38]. Wang Yu and Wang Yonghui investigated the overall level of tourism public services in Xinjiang from the perspective of tourists' perceptions to determine whether they meet tourism needs. The study shows that the highest level of tourist satisfaction is the perfect safety assurance mechanism [39]. From the viewpoint of the supply and demand relationship of industry development, ecosystem services play an important supporting role in rural tourism development [40]. Irena Travar concluded in her study that the most important features of destination image and service quality are the key strategic determinants that need to be identified first. Then, there is a need to ensure their preservation or to maintain a high level of service so that the commercialization of the destination does not compromise its quality [41].

Considering that the service quality aspect has been less explored in current research in the field of cultural tourism in Huizhou, this study attempts to focus on the factors influencing the service quality of cultural tourism in Huizhou, focusing on the analysis of public services such as tourism infrastructure, tourism transportation services, and convenient information services in the region, as well as exploring the correlation between the quality of the services and the local, regional economy. The TOPSIS method was used to empirically analyze the data in the Anhui Provincial Statistical Yearbook, and the coordination between rural cultural tourism and service quality was explored by mining the relationship between the refined indicator variables. The entropy weight TOPSIS method has been widely used in multi-attribute decision analysis due to its advantages of objectivity, high information utilization, and wide applicability. It overcomes the limitations of some traditional methods and provides a more scientific and rational decision-support tool. This study provides assistance in realizing the high-quality development of cultural and tourism services in the Huizhou region. It is specifically embodied in the aspects of improving tourists' experience, enhancing the competitiveness of the tourism industry, promoting the cultural heritage and development of the Huizhou region, and enhancing the image visibility of the Huizhou region so as to promote the sustainable development of the tourism industry in Huizhou and to realize the fusion of culture and tourism in the Huizhou region.

## 2. Methodology

### 2.1. Data Sources

This study evaluates the effective supply of public service quality, cultural service quality, and tourism industry performance in the Huizhou region according to entropy weight TOPSIS, respectively. The specific methodological process is as follows: first, data collection and processing. The statistical data of Anhui Provincial Statistical Yearbook (2019–2021), Huangshan Municipal Statistical Bulletin (2019–2021), Xuancheng Municipal Statistical Bulletin (2019–2021), and the people's governments of counties in 2019–2021 were collected and processed. Second, the comprehensive evaluation of the Yangtze River Basin was performed by entropy weight-TOPSIS. SPSSAU was used to calculate the weights and analyze the degree of proximity between the level of transverse development and the optimal scenario among the three districts and five counties in the Huizhou region. To study the longitudinal changes in Huizhou region from 2019 to 2022. Thirdly, using the spatial analysis function of ArcGIS (10.8) software, the sub-indicators and

comprehensive indicators of the three districts and five counties in Huizhou region are classified into three grades, low, medium, and high, and then evaluated and analyzed the spatial various levels of cultural and tourism service quality in the Huizhou region.

*2.2. Research Methods*

In order to assess the elements affecting the quality of cultural and tourism services in 3 districts and five counties in the Huizhou region, the entropy weight method and TOPSIS method are both extensively used. In competitiveness evaluation research, the TOPSIS and entropy weight methods are frequently coupled. Initially, C. E. Shannon created the entropy weight method, which is a highly objective valuation method that determines the information entropy of the index in order to quantify the valid information contained in known data and index weights. The approximation ideal solution ranking, or TOPSIS, is a multi-objective decision-making technique. The fundamental tenet is to specify the ideal and negative ideal solutions to the choice problem, compare them to the attainable answer, and then decide which is best. First, using the entropy weight method, the evaluation index weights of the factors affecting the quality of cultural and tourism services in 3 districts and five counties of the Huizhou region are calculated. Next, their comprehensive scores are determined and analyzed using the TOPSIS method.

The use of entropy weighting enhances the objectivity of determining the weights of indicators and significantly reduces the interference of subjective human judgment. The core of the method is to utilize the degree of variability of the data itself to assign weights: the smaller the degree of variability of the indicator, the lower its weight; conversely, the higher the weight. Although the entropy weight method provides an objective basis for weight calculation, in practical application, if the weight results are not reasonable, it can be combined with other methods, such as the hierarchical analysis method, to carry out manual intervention and adjustment in order to ensure the reasonableness of the evaluation results. In terms of data processing, the entropy weight TOPSIS method has the ability to normalize the data, which enables it to cope with different types and scales of data, thus enhancing the applicability and flexibility of the method. Compared with the hierarchical analysis method (AHP), which requires the construction of judgment matrices, the entropy-weighted TOPSIS method performs better in reducing subjectivity because the judgment matrices in the AHP are often affected by the decision maker's subjective preferences. In addition, the entropy weight TOPSIS method is particularly suitable for complex decision problems with multiple evaluation metrics, and the number of evaluation factors can be more than 10, which is different from the AHP, which may be limited when dealing with a larger number of decision factors. The entropy weight TOPSIS method has been widely used in multi-attribute decision analysis because of its strong objectivity, high information utilization, and wide applicability. It overcomes the limitations of some traditional methods and provides a more scientific and rational decision-support tool. Therefore, I choose to use this method of entropy weight TOPSIS to study the factors that influence cultural and tourism service quality in the Huizhou area.

2.2.1. Calculation of Indicator Weights Using Entropy Weight Method

In the first step, the resulting data are standardized, and since the dimensions of the values in the matrix are inconsistent, for each quantitative indicator, it is necessary to eliminate the differences in the magnitude of the original data.

$$G_{ij} = \frac{X_{ij}}{\sum_{i=1}^{m} X_{ij}} \tag{1}$$

In the formula, $i = 1, 2, \ldots, m; j = 1, 2, \ldots, n$. $x_{ij}$ is the raw data of the jth indicator in the ith district of Huizhou; $G_{ij}$ is the normalized value of $x_{ij}$.

In the second step, information entropy $Z_j$ is calculated:

$$Z_j = \frac{\sum_{i=1}^{m} G_{ij} \ln G_{ij}}{\ln m} \tag{2}$$

In the formula, m is the number of indicators.

Step 3, Calculate objective weights $W_j$ :

$$W_j = \frac{1 - Z_j}{\sum_{j=1}^{m} 1 - Z_j} \tag{3}$$

### 2.2.2. Calculate the Comprehensive Evaluation Score Using TOPSIS Method

The first step is to construct the standardization matrix $A = \left(X_{ijk}\right)_{m \times n \times k}$. Normalize the matrix of factors affecting the service quality of Huizhou cultural tourism and get the standardization matrix, $B = \left(b_{ijk}\right)_{m \times n \times k'}$ the normalization formula is as follows:

$$b_{ijk} = \frac{a_{ijk}}{\sum_{i=1}^{mk} a_{ijk}} \tag{4}$$

In the second step, the weighted normalization matrix is constructed $B = \left(b_{ijk}\right)_{m \times n \times k}$. The weighted normalization matrix is weighted to obtain the weighted normalization matrix $R = \left(r_{ij}\right)_{m \times n \times k}$. In the formula, $r_{ijk=W_j} b_{ijk}$. $W_j$ is the weight of the indicator determined by the entropy weighting method described previously.

The third step is to determine the set of positive and negative ideal solutions. Take the maximum value of the positive indicator and the minimum value of the inverse indicator to form the positive ideal solution set $R^+$, and take the minimum value of the inverse indicator and the maximum value of the positive indicator to form the negative ideal solution set $R^-$.

$$R^+ = (r_1^+, r_2^+, ..., r_n^+)$$
$$R^- = (r_1^-, r_2^{+-}, ..., r_n^-) \tag{5}$$

In $R^+$, $r_j^+ = (\max r_{ij}^+, \min r_{\bar{i}j})$. In the formula, $\max r_{ij}^+$ is the maximum value of the positive indicator j, $\min r_{\bar{i}j}$ is the inverse indicator j of the minimal value; In $R^-$, $r_j^- = (\min r_{ij}^+, \max r_{\bar{i}j})$. In the formula, $\min r_{ij}^+$ is the minimal value of the positive indicator j, $\max r_{\bar{i}j}$ is the inverse indicator j of the maximum value, $(j = 1, 2, ..., n)$.

In the fourth step, the relative closeness of the evaluation value of tourism service quality $C_{ik}$ is calculated.

$$C_{ik} = \frac{d_{ik}^-}{d_{ik}^+ + d_{ik}^-} \tag{6}$$

In the fifth step, the evaluation $C_{ik}$ results of the high-quality development of tourism in each region of Huizhou are ranked according to the size of the value of $C_{ik}$, and the closeness to the value has a decisive role in the high-quality development level of tourism in the region. The larger the value of $C_{ik}$ is, the closer the positive ideal solution is to the degree of excellent development of regional tourism, indicating that the degree of upscale tourism growth in the area is higher; the smaller the value of $C_{ik}$ is, the closer the level of high-quality development of regional tourism is to the negative ideal solution, indicating that the degree of a high-quality tourism development in the region is lower.

## 3. Study Area

The study area is the Huizhou region (Figure 1). The Huizhou region is the predecessor of the present-day Huangshan City, with Huizhou as its prototype, and also includes some of the neighboring pan-Huizhou regions and is located in the combination of Anhui, Zhejiang, and Gan provinces [42]. The traditional Huizhou region refers to the ancient Huizhou area consisting of Taiping County, Shexian County, Xiuning, Wuyuan,

Qimen, Yixian County, and Jixi. Its administrative map is relatively stable, with Wuyuan belonging to Jiangxi Province and the rest of the region belonging to Anhui Province. As the seat of ancient Huizhou, the local area has many historical figures and celebrities through the ages, and the history and culture are very developed. Zhu Xi, Cheng Minzheng, Dai Zhen, Tao Xingzhi, and so on are historical celebrities who have had important influence in various fields. Huizhou culture consists of economic and cultural genres such as Xin'an Science, Xin'an Medicine, Anhui Opera, Hui Printmaking, Hui Seal Engraving, Hui Bonsai, etc., which are profound and have a long history [43]. Huizhou District is located in the central part of Huangshan City. Its Fengle River is the "mother river" of Huizhou District and belongs to the subtropical monsoon humid climate zone; the average rainfall for many years is 1728.2 mm, but the inter-annual distribution of uneven, spring rainy, rainy, autumn drought, winter rainfall [44]. The region has seven counties and two cities under its jurisdiction, with a total area of 1.34 million square kilometers and a total population of 1.79 million. The administrative office was initially located in Shexian County and later moved to Tunxi, which is geographically located between longitude 118°04′10″–118°53′50″ E and latitude 29°30′25″–30°09′10″ N. The region has a unique historical background and rich historical and cultural relics, and there are many traditional ancient villages representing Huizhou [45], immersing tourists in the unique historical atmosphere of Huizhou and making Huizhou a popular tourist region today.

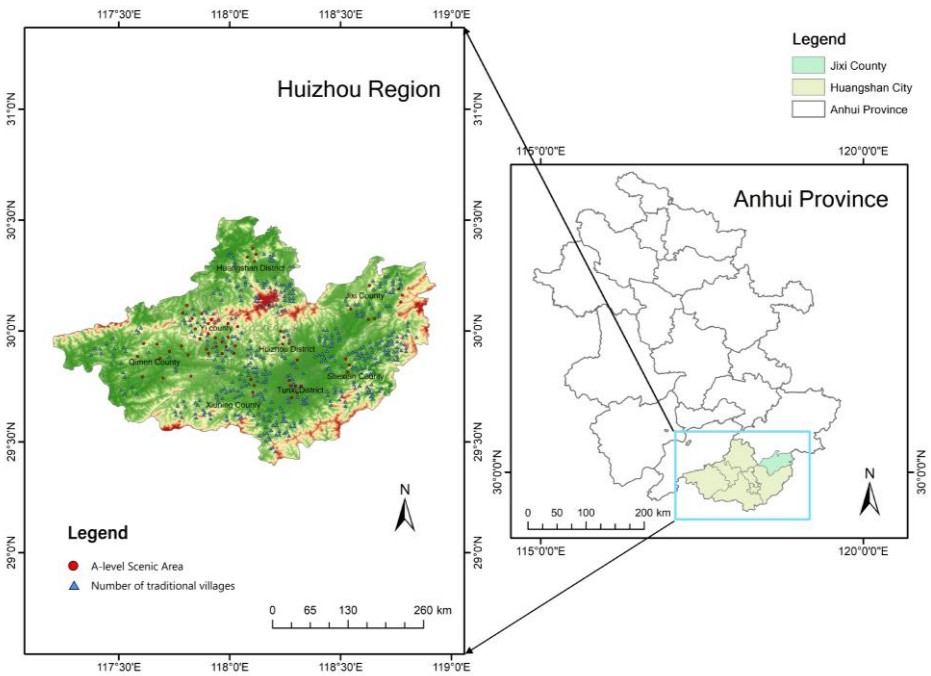

**Figure 1.** Study area of Anhui Huizhou region.

The culture of Huizhou includes not only the intangible cultural heritage and folk customs of Huizhou but also the living habits and lifestyles of the aborigines in the traditional villages of Huizhou, which have been formed for thousands of years and constitute the unique regional culture of the area. Ideologically, Huizhou is the birthplace of the Cheng-Zhu school of reasoning; in terms of painting and sculpture, Huizhou promotes the Xin'an School of Painting and the Huizhou School of Sculpture. The migration of the Central Plains lineage created an influential ethnic society where man and nature developed in harmony with a strong humanistic flavor. The local economy of Huizhou is spiritually driven by its culture, which also serves as the region's spiritual foundation. Meanwhile, Culture and tourism go hand in hand. Since the 1980s, when it

first became a recognized professional concept, discussions about tourism culture have frequently featured in newspapers, magazines, and even certain literary works. Many academics and industry professionals have long accepted the notion that "culture is the soul of tourism" when interpreting the connection between the two. In essence, visitors primarily consume culture, and culture is where they start and where they end up when they travel. Tourists participate in specific projects, such as enjoying cliff carving, watching folk songs and dances, etc., which is the main content of tourism activities. So, tourism itself is a cultural activity. The distinctive "intangible cultural heritage" of Huizhou draws in a lot of tourists, and in order to integrate culture and tourism, Huizhou culture must be creatively transformed and innovatively developed while taking tourist preferences into account. Prerequisites include the preservation of local ecosystems and regional culture. Intangible cultural heritage, as a tool for cultural exchange, not only attracts tourists but also has great significance in enhancing cultural confidence and national identity. Huizhou region is rich in tourism resources. According to statistics, there are currently 61 A-class picturesque locations, including 14 5A-class locations. Xidi Hongcun, Huangshan Mountain Scenic Area, and Chengkan are popular attractions in Huizhou, receiving tens of millions of visitors every year. In addition, Huizhou has many traditional Chinese ancient villages and historical cities and towns. The ancient city of Shexian County is a national historical and cultural city, is the four treasures of Huizhou ink, the main production area of Sheyan ink stone, the cradle of the national essence of Peking Opera, the main birthplace of Huizhou merchants. The connotation of Huizhou culture is the ecological consciousness of "the unity of heaven and mankind," and its structure is influenced by the natural environment, such as mountains, waters, temperature, and other human factors such as architecture, feng shui, and economic behavior. The ancient Huizhou people revered nature and sought to unite man, nature, and architecture when they constructed traditional communities. The outcome was that "mountains, screens, and water" provided the atmosphere in which thousands of Huizhou traditional villages grew up. An organic cultural ecosystem was formed, integrating human beings, villages, and the environment. Habitat patterns, architectural forms, and social relations formed during long-term development have profound ecological connotations. The United Nations Educational, Scientific, and Cultural Organization (UNESCO) inscribed Xidi and Hongcun as a World Heritage Site in 2000. Here, there are two adjacent historical villages in South Anhui, only 40 km away from the Huangshan Mountain Scenic Spot, that are famous for their well-preserved Huizhou-style architecture. Xidi Village is known as the "Museum of Ming and Qing Dynasty Local Houses", and Hongcun Village is known as "China's Beautiful Painted Countryside".

## 4. Indicator System Establishment

Based on social, economic, environmental, and resource influencing factors, this paper selects 20 evaluation indicators from three aspects, namely public service quality, cultural service quality, and tourism industry performance, to assess the international influence of tourism, the economic contribution of tourism, and the results of open development. The indicator system is constructed to contain three levels: target level, guideline level, and indicator level (Table 1). In Table 1, "+" represents the entire indicator, positive indicators refer to indicators that increase in value as the phenomenon becomes better or more severe. On the contrary, "-" represents a negative indicator, which refers to an indicator that increases in value as the phenomenon becomes more severe or unfavorable. Among them, public service quality (B1) refers to the public welfare services provided by tourism destinations to tourists, such as tourism transportation services, tourism convenience and benefits services, tourism safety and security services, and tourism public information services. Cultural service quality (B2) refers to the fact that cultural services are behaviors that satisfy people's cultural interests and needs. Tourism Industry Performance (B3) reflects the economic development outcomes and relevant effects of tourism.

Quality of public services B1: The quality of public services determines the potential attractiveness of tourism. The quality of public services represents the level of tourism infrastructure development, and seven indicators have been developed on the basis of assessing the level of synergistic development of tourism and other related industries. Among them, tourism-related industries mainly include accommodation, catering, and transportation, and six positive indicators and one negative indicator are established. The increase in the mileage of roads (C1) implies the improvement of the transportation network, which allows tourists to reach various tourist attractions more conveniently, shortening the travel time and improving the tourism experience. The number of star-rated hotels (C3), five-star hotels (C4), and well-known travel agencies (C5) usually have higher service standards and management levels and are able to provide tourists with better and more professional services, thus improving the quality of public services in the region. The number (C2) and distribution of public restrooms determine the convenience of residents and tourists in using them. Especially in crowded areas such as tourist attractions, commercial districts, and transportation hubs, adequate public toilets can greatly enhance people's travel comfort. Commercial facilities (C6) such as cafes, restaurants, and movie theaters not only provide services to local residents but also attract tourists and promote tourism, which is also part of enhancing the quality of regional public services. The situation of tourism complaints (C7) is one of the key indicators of tourist satisfaction. If there is an increase in the number of complaints, it may indicate that tourists' expectations are not being met, which affects their overall evaluation of the destination.

Cultural service quality B2: The main purpose of cultural tourism is to provide tourists with a learning environment and an appreciation of the culture of each place. Tourists can gain knowledge, satisfaction, and emotional needs in the process. Tourism needs cultural additions, which will enhance the cultural identity of the tourism region. Seven indicators were set up to analyze the factors that influence cultural tourism services in the Huizhou region. These indicators reflect the richness of cultural resources in the Huizhou region. The number of museums (C12) and the number of art performance venues (C13), as the core elements of cultural tourism, play a key role in enhancing the region's tourism attractiveness. Tourists are often attracted to these locations with educational significance and aesthetic value. An increase in the number of intangible cultural heritages (C8) can further enrich the cultural connotations of tourist attractions, resorts, and other tourist spaces and enhance the cultural heritage. The number of historical and cultural cities and towns (C9) and the number of traditional villages (C14) are important carriers of traditional culture, possessing rich historical relics, ancient buildings, traditional styles, etc. These unique cultural resources cannot be copied by other regions and can attract a large number of tourists to come to visit them. The 5A scenic spots (C10) and the A-level scenic spots (C11) often contain rich cultural heritage, and the development of tourism and the development of the A-level scenic spots can revitalize historical resources and make them play a new cultural value in modern society. The higher the number of the above indicators, the richer the cultural resources in Huizhou, and the cultural connotation is the foundation of tourism. In the current context of the integration of culture and tourism, the richer the cultural resources, the higher the sustainability of tourism.

Tourism Industry Performance B3: Six indicators were established to reflect the economic benefits brought by tourism in the Huizhou region. With policy support, the digital platform of cultural tourism in the Huizhou region has been improved, attracting more tourists from China and other countries. Measuring the economic performance of the tourism industry bridges the gap in the study of the economic benefits of cultural tourism in the Huizhou region. The tourism industry performance is the final result of tourism development, and tourism industry performance research has always been an important part of the tourism research field. The better the development of the tourism industry in the Huizhou region, the higher the economic benefits it brings, and the better

the tourism industry's performance. The increase in the number of people received (C15), the number of overseas tourists received (C16), and the number of domestic tourists (C17) can lead to an increase in the direct income of tourism-related industries such as tickets, food, and beverage, and lodging, among which the increase in the number of overseas tourists received enhances the international popularity of the destination, and the international cooperation has become an important way to enhance the performance of the tourism industry. Gross annual revenue from tourism (C18) and foreign exchange earnings from tourism (C19) are key economic indicators for measuring the performance of a region's tourism industry. These two indicators not only directly reflect the economic efficiency of a tourist destination but also the overall level of tourism services and international competitiveness of the region. The Gross Accommodation and Catering Product (C20) is a key economic indicator for measuring the performance of a region's tourism industry, which directly reflects the economic benefits of the tourism industry in terms of accommodation and catering, leading to the extension of the related industry chain and the formation of a complete tourism industry system.

**Table 1.** Indicators of factors affecting the quality of cultural and tourism services in the Huizhou area.

| Target Level | Standardized Layer | Program Level | Unit | Direction of Indicators |
|---|---|---|---|---|
| A: Research on the Influencing Factors of Cultural and Tourism Service Quality in Huizhou Area | B1 Quality of public services | C1 road mileage | kilometers | + |
| | | C2 Number of public toilets | one | + |
| | | C3 Number of star hotels | one | + |
| | | C4 Number of five-star hotels | one | + |
| | | C5 Number of travel agencies | one | + |
| | | C6 Number of commercial facilities | one | + |
| | | C7 Complaints about tourism | one | − |
| | B2 Quality of cultural services | C8 Number of intangible cultural heritages | one | + |
| | | C9 Number of historical and cultural cities and towns | one | + |
| | | C10 Number of 5A-level tourist attractions | one | + |
| | | C11 Number of A-class tourist attractions | one | + |
| | | C12 Number of museums | one | + |
| | | C13 Performing arts venue | one | + |
| | | C14 Traditional village | one | + |
| | B3 Tourism industry performance | C15 Number of persons received | Ten thousand people | + |
| | | C16 Reception of overseas visitors | Ten thousand people | + |
| | | C17 Number of domestic tourists | Ten thousand people | + |
| | | C18 Gross annual income from tourism | billions | + |
| | | C19 Foreign exchange earnings from tourism | Ten thousand dollars | + |
| | | C20 Gross Domestic Product (GDP) | Ten thousand dollars | + |

## 5. Results

### 5.1. Intercomparison of Different Indicators

The table shows the collection of data on indicators for 2019–2021 (Tables 2–4), utilizing the entropy weight method to calculate the weight value of each indicator of tourism service quality in 3 prefectures and five counties of Huizhou region from 2019–2021, as well as the weight value of sub-indicators such as public service quality, cultural service quality, and tourism industry performance (Table 5). Table 2 shows the sub-indicators of the elements affecting the quality of tourism services in the three prefectures and five counties of the Huizhou region, which are the number of 5-star hotels, the number of locations for artistic performances, and the number of traditional villages. The average weights of these three factors over the course of three years are all 0.0923.

**Table 2.** Data Collection for the Study of Factors Influencing the Quality of Cultural and Tourism Services in Huizhou Region in 2019.

|     | H1     | H2     | H3      | H4      | H5    | H6      | H7     | H8     |
|-----|--------|--------|---------|---------|-------|---------|--------|--------|
| C1  | 239    | 381    | 1200    | 2023    | 1532  | 626     | 1314   | 210    |
| C2  | 27     | 39     | 35      | 99      | 7     | 69      | 23     | 63     |
| C3  | 10     | 1      | 8       | 4       | 0     | 3       | 1      | 1      |
| C4  | 3      | 1      | 0       | 0       | 0     | 0       | 1      | 0      |
| C5  | 115    | 13     | 46      | 43      | 24    | 43      | 17     | 21     |
| C6  | 12     | 1      | 7       | 5       | 2     | 3       | 1      | 2      |
| C7  | 220    | 33     | 69      | 64      | 17    | 14      | 5      | 4      |
| C8  | 49     | 69     | 26      | 28      | 33    | 79      | 20     | 21     |
| C9  | 9      | 4      | 3       | 7       | 6     | 7       | 3      | 1      |
| C10 | 4      | 3      | 2       | 2       | 0     | 2       | 0      | 1      |
| C11 | 6      | 6      | 4       | 3       | 3     | 22      | 10     | 7      |
| C12 | 17     | 4      | 5       | 6       | 6     | 3       | 5      | 5      |
| C13 | 6      | 20     | 17      | 10      | 4     | 1       | 15     | 3      |
| C14 | 14     | 9      | 74      | 130     | 149   | 44      | 28     | 28     |
| C15 | 1478   | 678.12 | 1350.58 | 1142.12 | 686   | 1376.58 | 340.37 | 447.1  |
| C16 | 70.1   | 27.78  | 55.99   | 35.8    | 24.1  | 53.8    | 0.86   | 21.8   |
| C17 | 1407.9 | 650.34 | 1294.59 | 1106.3  | 661.4 | 1322.8  | 340.4  | 425.3  |
| C18 | 173.7  | 46.3   | 11.47   | 103.3   | 55.2  | 114.76  | 23.71  | 28.6   |
| C19 | 31,602 | 9148   | 13,262  | 11,479  | 7523  | 13,586  | 131    | 125    |
| C20 | 84,985 | 13,540 | 73,166  | 44,381  | 19,020| 24,353  | 17,460 | 19,325 |

**Table 3.** Data Collection for the Study of Factors Influencing the Quality of Cultural and Tourism Services in Huizhou Region in 2020.

|     | H1   | H2   | H3   | H4   | H5   | H6   | H7   | H8    |
|-----|------|------|------|------|------|------|------|-------|
| C1  | 232  | 426  | 1389 | 2065 | 1625 | 621  | 1305 | 213.4 |
| C2  | 27   | 40   | 35   | 99   | 8    | 69   | 23   | 63    |
| C3  | 10   | 1    | 8    | 3    | 0    | 3    | 1    | 1     |
| C4  | 3    | 1    | 0    | 0    | 0    | 0    | 1    | 0     |
| C5  | 110  | 10   | 44   | 45   | 24   | 41   | 17   | 21    |
| C6  | 10   | 1    | 7    | 5    | 2    | 2    | 1    | 2     |
| C7  | 199  | 26   | 66   | 60   | 13   | 19   | 5    | 5     |
| C8  | 50   | 70   | 26   | 30   | 33   | 79   | 20   | 21    |
| C9  | 9    | 4    | 3    | 7    | 6    | 7    | 3    | 1     |
| C10 | 4    | 3    | 2    | 2    | 0    | 2    | 0    | 1     |
| C11 | 6    | 6    | 4    | 3    | 3    | 22   | 10   | 7     |
| C12 | 17   | 4    | 5    | 6    | 6    | 3    | 5    | 5     |

| | | | | | | | | |
|-----|-------|-------|--------|--------|--------|--------|--------|--------|
| C13 | 6 | 20 | 17 | 10 | 4 | 1 | 15 | 3 |
| C14 | 14 | 9 | 75 | 144 | 149 | 44 | 28 | 31 |
| C15 | 917 | 385.6 | 763.3 | 662.98 | 406.1 | 874.72 | 197.4 | 447.1 |
| C16 | 13.7 | 4.9 | 10 | 8.29 | 4.4 | 11.2 | 0.2 | 21.8 |
| C17 | 903.3 | 380.7 | 753.3 | 654.69 | 950 | 2679 | 197.2 | 425.3 |
| C18 | 100.9 | 22.96 | 54.1 | 55.01 | 31.6 | 68.8 | 12.3 | 28.6 |
| C19 | 5727 | 1372 | 2300 | 1722 | 950 | 2679 | 26 | 19 |
| C20 | 69,888 | 11,276 | 65,722 | 41,190 | 16,886 | 21,665 | 15,694 | 17,325 |

**Table 4.** Data Collection for the Study of Factors Influencing the Quality of Cultural and Tourism Services in Huizhou Region in 2021.

| | **H1** | **H2** | **H3** | **H4** | **H5** | **H6** | **H7** | **H8** |
|-----|--------|--------|--------|--------|--------|--------|--------|--------|
| C1 | 228 | 426 | 1393 | 2066 | 1622 | 620 | 1317 | 218.3 |
| C2 | 27 | 43 | 35 | 99 | 8 | 69 | 23 | 64 |
| C3 | 10 | 1 | 8 | 3 | 0 | 3 | 1 | 1 |
| C4 | 3 | 1 | 0 | 0 | 0 | 2 | 1 | 0 |
| C5 | 110 | 8 | 44 | 45 | 24 | 40 | 17 | 21 |
| C6 | 10 | 1 | 7 | 5 | 2 | 2 | 1 | 2 |
| C7 | 255 | 25 | 42 | 8 | 27 | 7 | 7 | 10 |
| C8 | 53 | 73 | 26 | 33 | 36 | 81 | 20 | 23 |
| C9 | 9 | 4 | 3 | 7 | 6 | 7 | 3 | 1 |
| C10 | 4 | 3 | 2 | 2 | 0 | 2 | 0 | 1 |
| C11 | 6 | 6 | 4 | 3 | 3 | 22 | 10 | 7 |
| C12 | 17 | 4 | 5 | 6 | 6 | 6 | 3 | 5 |
| C13 | 6 | 20 | 17 | 10 | 5 | 1 | 15 | 3 |
| C14 | 14 | 11 | 79 | 148 | 153 | 46 | 25 | 31 |
| C15 | 1344.2 | 566.52 | 1116.9 | 969.9 | 592.12 | 1273.2 | 285.9 | 319.9 |
| C16 | 0.88 | 0.377 | 0.57 | 0.64 | 0.33 | 0.91 | 0.03 | 0.01 |
| C17 | 1343.32 | 566.143 | 1116.33 | 969.3 | 591.8 | 1272.3 | 285.9 | 319.8 |
| C18 | 154.3 | 34.69 | 81.8 | 82.6 | 46.9 | 101.6 | 18.1 | 30.86 |
| C19 | 210.2 | 70.5 | 90.9 | 103.1 | 80.5 | 135.5 | 0.9 | 0.5 |
| C20 | 62,705 | 11,178 | 60,413 | 40,680 | 17,552 | 22,586 | 16,361 | 18,235 |

(1) The factor determining the quality of culture and tourism services in the three provinces and five counties of the Huizhou region with the most weight is the public service quality sub-indicator. In the tourism sector, the idea of public service quality is defined using the expectation-perception theory, which focuses mostly on urban tourist sites. The impact of public service quality on rural tourist growth is shown by the robust development of rural tourism practices, but the differences in tourism infrastructure, tourism transportation, tourism information service, tourism public environment, tourism safety and security, and tourism culture and entertainment make the level of public service of rural tourism show differences in different geographic regions and spaces. Therefore, it makes sense that among the variables affecting the standard of cultural and tourism services in Huizhou's three prefectures and five counties, the public service quality indicator has the highest weight value. Improving the public service system for tourism is therefore crucial to the advancement of Huizhou's tourism service quality.

(2) The weight of cultural service quality indicators in the cultural and tourism service quality influencing factors of the three prefectures and five counties in Huizhou is in second place. The growth of rural tourism in China's tourist sector is accelerating, and it is this sector's ice-breaker industry that effectively promotes the opening of other industries. The Huizhou region has been at the forefront of tourism in the Anhui region

because of its unique regional culture. Huizhou attracts tourists from all over the country with its long history of cultural connotation and simple folk customs, contributing to the growth of tourism in the province of Anhui and even in China, and its tourism revenue has long been in the forefront of Anhui Province. Therefore, the cultural service quality sub-indicator has a larger weight value among the factors influencing the quality of cultural and tourism services in the three prefectures and five counties of Huizhou region.

**Table 5.** Information entropy and weights of indicators.

| Indicator Layer | $E_i$ | $W_i$ |
|:---:|:---:|:---:|
| C1 | 0.3437 | 0.0526 |
| C2 | 0.4488 | 0.0348 |
| C3 | 0.9006 | 0.0576 |
| C4 | 0.9006 | 0.0894 |
| C5 | 0.5156 | 0.0489 |
| C6 | 0.9006 | 0.0715 |
| C7 | 0.3513 | 0.0159 |
| C8 | 0.5424 | 0.0549 |
| C9 | 0.0000 | 0.0288 |
| C10 | 0.0000 | 0.0423 |
| C11 | 0.0000 | 0.0759 |
| C12 | 0.9006 | 0.0592 |
| C13 | 0.9006 | 0.0396 |
| C14 | 0.3700 | 0.0640 |
| C15 | 0.3613 | 0.0424 |
| C16 | 0.5029 | 0.0389 |
| C17 | 0.3606 | 0.0424 |
| C18 | 0.3437 | 0.0438 |
| C19 | 0.5796 | 0.0413 |
| C20 | 0.5301 | 0.0556 |

*5.2. Horizontal Comparison between Different Regions*

Combined with the entropy weighting results, the relative proximity of each region in the Yangtze River Basin in 2019, 2020, and 2021 was calculated (Table 6). The distribution map of evaluation grades for each region in Huizhou throughout the study period is created using ArcGIS software(10.8) using the relative proximity index of each region separated into four grades (Figure 2). Data on the variables influencing the quality of cultural and tourism services in Huizhou in 2019–2021 were used to gauge the development of high–quality tourism, and the data were analyzed from the perspectives of spatial differences and temporal evolution, respectively.

Three districts and counties, Tunxi District (H1), Yixian County (H6), and Huangshan District (H3), are at the top of the score value of tourism development level in Huizhou. Most of the popular scenic spots in Huizhou are located in Tunxi District and Huangshan District, which has a location advantage. Moreover, as the central urban area with the highest level of economic development in Huangshan City, Tunxi District does not have a high dependence on tourism compared to other districts and counties. However, Tunxi District has more commercial and recreational facilities, which attract a large number of passengers and greatly promote the development and upgrading of the tourism industry in Huizhou. Yixian County's public service quality indicators are not excellent, but each of them is in the middle and upper levels of the Huizhou region, with a good balance. There is not much difference with Tunxi District, thanks to Yixian County's superiority of its natural environment and the richness of tourism. According to statistical data, Yixian County has 22 A-level scenic spots as of 2021, including two 5A-level scenic spots, which

is far more than other districts and counties in Huizhou region, and 81 intangible cultural heritages, which accounts for the largest proportion of intangible cultural heritages in Huizhou region. It can be seen that the quality of cultural services has a greater impact on the tourism industry in Huizhou, and Yixian County has given full play to the many local tourist attractions and unique regional culture and has gained an advantage in the local tourism industry, which in turn has led to the development of the local economy in Yixian County. Huangshan District occupies the largest area in the whole Huizhou region, and Huangshan County is not ranked at the top of all indicators, but it is in the middle and upper level in the Huangzhou region, and it is well-balanced.

Shexian County (H4), Huizhou District (H2), and Xiuning County (H5) rank in the middle. In this case, the index value of Xiuning County lags behind that of Huangshan District and Tunxi District, which are also at the forefront of the tourism industry, although the quality of cultural services in Huizhou District is close to other districts and counties with high scoring values, the quality of public services is lower than that of Yixian County, which has a similar size of administrative jurisdiction; Shexian County is a national historical and cultural city, and was the political, economic and cultural center of ancient Huizhou. Shexian County is the birthplace of Huizhou culture and the main birthplace of Huizhou merchants and Huizhou cuisine. Therefore, although Shexian County is a tourist city with tourism as its leading industry, its comprehensive index score is low and is at a medium level in the province. The remaining score values of Qimen County (H7) and Jixi County (H8) rank at the bottom. The two counties of Qimen County and Jixi County have significantly lower values of each indicator than Yixian County and Huangshan District, which are rich in tourism resources, and Tunxi District, which is economically developed. Jixi County, located in Xuancheng City, is significantly affected by the size of its administrative district and the level of economic development, reflecting a lower competitiveness of tourism.

**Table 6.** The 2019–2021 response of traditional village tourism in the Huizhou area

| | $d_j^+$ | | | $d_j^-$ | | | Total Relative Proximity Value | | | Ranking |
|---|---|---|---|---|---|---|---|---|---|---|
| | **2019** | **2020** | **2021** | **2019** | **2020** | **2021** | **2019** | **2020** | **2021** | |
| H1 | 0.108 | 0.115 | 0.115 | 0.207 | 0.199 | 0.195 | 0.658 | 0.634 | 0.628 | 1 |
| H2 | 0.193 | 0.195 | 0.189 | 0.087 | 0.086 | 0.085 | 0.312 | 0.306 | 0.309 | 5 |
| H3 | 0.179 | 0.177 | 0.156 | 0.112 | 0.112 | 0.123 | 0.385 | 0.387 | 0.441 | 3 |
| H4 | 0.179 | 0.182 | 0.158 | 0.109 | 0.108 | 0.124 | 0.378 | 0.372 | 0.439 | 4 |
| H5 | 0.209 | 0.209 | 0.194 | 0.077 | 0.078 | 0.088 | 0.269 | 0.272 | 0.311 | 6 |
| H6 | 0.18 | 0.18 | 0.136 | 0.123 | 0.127 | 0.146 | 0.406 | 0.413 | 0.518 | 2 |
| H7 | 0.202 | 0.201 | 0.201 | 0.067 | 0.067 | 0.063 | 0.249 | 0.249 | 0.239 | 7 |
| H8 | 0.222 | 0.218 | 0.212 | 0.037 | 0.049 | 0.038 | 0.141 | 0.182 | 0.153 | 8 |

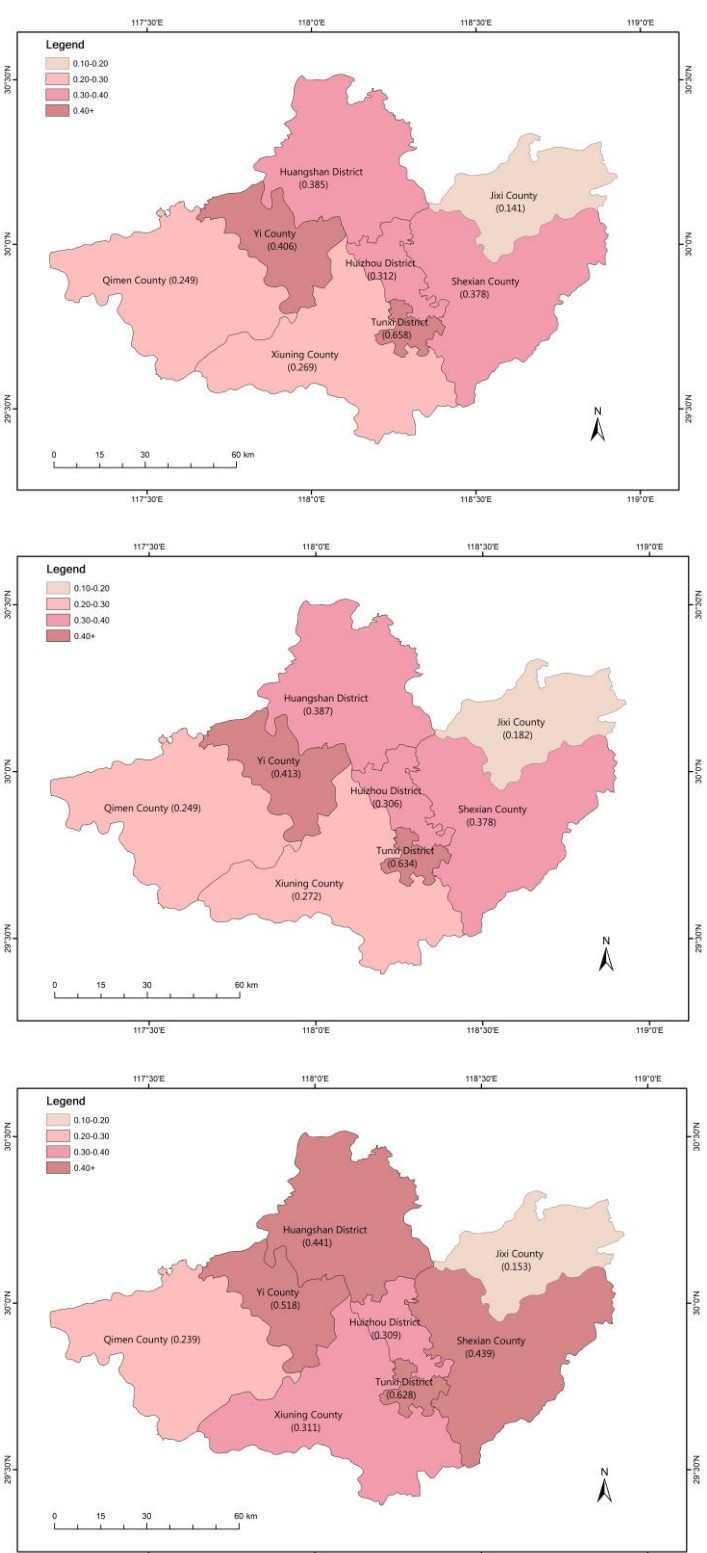

**Figure 2.** Distribution of evaluation ratings in the Emblem area, 2019–2021.

### 5.3. Longitudinal Comparison of Different Years

Combining the entropy weighting results, the relative proximity of the indicators in the Huizhou region is calculated for the three years of 2019, 2020, and 2021 (Table 7). In the three years from 2019 to 2021, the Huizhou region's tourism sector exhibits an ebb-and-flow pattern. The quality of the region's public and cultural services doesn't vary significantly, but the sector's performance does (Table 8). From 2019 to 2020, the tourism

industry performance ushered in a substantial decline, with the total number of people received falling from 74.98 million to 46.54 million, a year-on-year decline of 37.93%, and the number of overseas tourists falling from 2.923 million to 744,900, a year-on-year decline of 74.33%; from 2020 to 2021, the total number of people received rose from 46.54 million to 64.68 million, a year-on-year rise of 38.98%, and the number of overseas tourists dropped from 744,900 to 37,470,000, a year-on-year decline of 94.97%. The unpredictable variation in tourism income is caused by the sharp fluctuation in the number of visitors. The total annual income of tourism in the Huizhou region fell from 55.704 billion yuan in 2019 to 37.427 billion yuan in 2020, a year-on-year drop of 32.81%; the total annual income of tourism in 2020 rose from 37.427 billion yuan to 55.085 billion yuan in 2021, a year-on-year rise of 47.18%. In the past three years, although the number of tourists has been warming up, the number of overseas tourists has been sharply reduced, and tourism in the Huizhou region has suffered a major blow.

**Table 7.** The relative proximity of factors influencing cultural tourism in the Emblem region, 2019–2021.

|  | $d_j^+$ | $d_j^-$ | **Total Relative Proximity Value** | **Ranking** |
|---|---|---|---|---|
| 2019 | 0.186 | 0.182 | 0.493 | 2 |
| 2020 | 0.237 | 0.062 | 0.206 | 3 |
| 2021 | 0.175 | 0.191 | 0.522 | 1 |

**Table 8.** Collection of indicator data on cultural and tourism influencing factors in the Huizhou region from 2019 to 2021.

|  | **2019** | **2020** | **2021** |
|---|---|---|---|
| C1 | 7525 | 7876.4 | 7890.3 |
| C2 | 362 | 364 | 368 |
| C3 | 28 | 27 | 27 |
| C4 | 5 | 5 | 7 |
| C5 | 322 | 312 | 309 |
| C6 | 33 | 30 | 30 |
| C7 | 426 | 393 | 381 |
| C8 | 325 | 329 | 345 |
| C9 | 40 | 40 | 40 |
| C10 | 14 | 14 | 14 |
| C11 | 61 | 61 | 61 |
| C12 | 51 | 51 | 52 |
| C13 | 76 | 76 | 77 |
| C14 | 476 | 494 | 507 |
| C15 | 7498.87 | 4654.2 | 6468.64 |
| C16 | 290.23 | 74.49 | 3.747 |
| C17 | 7209.03 | 6943.49 | 6464.893 |
| C18 | 557.04 | 374.27 | 550.85 |
| C19 | 86,856 | 14,795 | 692.1 |
| C20 | 296,230 | 259,646 | 249,710 |

## 6. Discussion

There are significant spatial differences in tourism resource endowments in the Huizhou region, which are mainly reflected in the quality of public services and the quality of cultural services. Well-established public facilities and rich cultural resources are important indicators of the service quality of a tourist destination. First, the number of five-star hotels has the greatest impact on the quality of public services, and its increase can directly enhance the tourism image and attractiveness of Huizhou and attract high-



quality tourists who emphasize the accommodation experience. The government and enterprises should work together to guide and incentivize five-star hotels to continuously improve their service quality, as well as to create unique tourism products by combining local cultural characteristics to meet higher-level market demand. In the Huizhou region, 80% of the districts have medium and above resource endowments, showing a high overall endowment. Tunxi, Huizhou, and Huangshan districts in Huangshan City do not differ much in terms of the quality of cultural services, mainly in terms of the quality of public services. This indicates that high public service quality has significant advantages for rural tourism development. In addition, the public service quality of rural tourism indirectly affects tourist loyalty through the conduction effect of destination image [46,47]. Therefore, it is crucial to improve the quality of tourism services in the competitive tourism market [48]. High-quality public services significantly enhance the attractiveness of Huizhou as a tourist destination, such as clean streets, convenient transportation systems, and abundant entertainment options, which are important factors that attract tourists. Furthermore, quality services can also improve Huizhou's competitiveness in the global tourism market, attracting more domestic and international tourists to the region.

Secondly, the quality of cultural services in Huizhou indicates that local cultural resources have an important impact on tourism development. Among them, the number of A-grade scenic spots and the number of museums have the highest weights, indicating that these factors are most likely to influence the level of supply of quality cultural services in the Huizhou region. High-quality cultural services not only help to protect and pass on the cultural heritage of Huizhou but also enable tourists to better understand and respect the history and culture of Huizhou through proper cultural display and education, thus stimulating the vitality of the local culture so that it can be better preserved and passed on to future generations. At the same time, this also provides local residents with the opportunity to understand and be proud of their own culture and enhance their cultural confidence. There are significant differences in the quality of cultural services in Huizhou, with areas with high support including Tunxi District, Huizhou District, and Yixian County in Huangshan City; medium areas including Huangshan District, Shexian County, and Xiuning County; and low areas Qimen County and Jixi County. The proportion of the three is 37.5%, 37.5%, and 25%, respectively, indicating that the richer the cultural resources, the more advantageous the area is in tourism development, while the areas with fewer cultural resources constrain the development of the local rural tourism industry due to the lack of widespread dissemination of folklore and customs or the lack of good cultural promotion policies. Although Huizhou cultural tourism has achieved some economic benefits since the 1920s, it still remains in the overall primary stage of landscape and cultural tours with ancient villages as the main body [49]. The simple reproduction of these material cultures can hardly reflect the local characteristics and ideological essence of Huizhou culture and cannot satisfy tourists' comprehensive appreciation of Huizhou culture [50]. One of the determinants of service quality and tourist satisfaction is the "authenticity" of heritage tourism resources [51,52]. Therefore, the government should take measures to actively explore the local characteristics of culture, form cultural symbols, and fully play a role in the regional advantages of Huizhou so as to promote the development of tourism. It should strengthen the protection of Huizhou's cultural heritage, such as ancient buildings, documents, and works of art, and implement scientific protection measures to prevent damage to cultural resources caused by tourism development.

Third, the tourism industry performance in Huizhou reflects the comprehensive situation of the local tourism industry in terms of economic benefits and cost performance. Tourism industry performance is directly affected by the number of tourists and the level of consumption and indirectly by the quality of public services and cultural services. It is an important indicator of tourism economic performance. Among them, the number of overseas tourists received, and the Gross Domestic Product (GDP) of accommodation and food services carry the highest weight. Overseas tourists usually have high spending

power, and their expenditures on accommodation, catering, shopping, and entertainment in the course of tourism directly increase local tourism revenue. In addition, high-spending overseas tourists bring more economic benefits and have a direct impact on boosting regional GDP. Accommodation and catering are the main sources of economic benefits from Huizhou's cultural tourism. When tourists stay and dine in the Huizhou region, their consumption is directly transformed into income for the local economy, and this increase in income helps to enhance regional GDP and promote local economic development. The performance of the tourism industry in Huizhou region shows significant differences, with Tunxi District and Yixian County showing the best performance, Huangshan District and Shexian County medium, and Huizhou District, Xuning County, Qimen County and Jixi County lower. Areas with high service quality have better economic performance, and areas with scarce resources and low service quality have worse performance. The flow of international tourists and the importance of tourism to the economies of many countries has steadily increased over the past few decades [53]. Many governments have placed greater emphasis on supporting and promoting tourism as a potential source of economic growth and employment and as a sector that adds value to cultural, natural, and other capital in the absence of market prices [54,55]. Therefore, the unique cultural resources of the Huizhou region should be explored, and the quality of tourism services should be improved so as to enhance the performance of the tourism industry and drive the development of the local economy [56].

Fourth, a similar study, "Spatio-temporal Evolution and Causal Analysis of Rural Tourism Heat in Jilin Province Based on Multivariate Data," focuses on regional resource endowment and transportation conditions. The study points out that improving road connectivity significantly promotes rural prosperity, and the development of rural roads facilitates mobility between urban and rural areas, promotes the development of impoverished and underdeveloped areas, and creates conditions for rural tourism. With the growth of pan-tourism, the development of tourism resources has deepened, allowing the industry chain to expand. This trend enhances the influence of rural tourism and brings new opportunities. This study is instructive for the improvement of rural tourism resources, industrial integration, and sustainable development, and it helps to understand the layout of regional rural tourism development.

## 7. Conclusions

This study developed a set of indicator systems for assessing the factors affecting the quality of cultural and tourism services, designing 20 indicators from three dimensions: public service quality, cultural service quality, and tourism industry performance. The weights of the indicators and their relationship to the level of tourism development in each location were also determined using the entropy weight TOPSIS approach through an empirical investigation of the variables that influence the caliber of tourism services. The results of the analysis of this proximity and weights are as follows (in order): The number of five-star hotels, the number of A-grade scenic locations, and the number of foreign visitors received are the top three sub-indicators of the elements impacting the quality of culture and tourism services in the Huizhou region. In the Huizhou region, the development of cultural resources and the improvement of public services are two important influencing factors that have an impact on the growth of tourism service quality. The top three districts and counties of Huizhou region in the comprehensive ranking are Tunxi District, Yixian County, and Huangshan District; the districts and counties with middle ranking are Shexian County, Xiuning County, and Huizhou District, and the districts and counties with lower ranking are Qimen County and Jixi County. Among them, the Tunxi District has the highest quality of tourism services, the highest number of tourists, and the highest economic benefits of tourism, and is perennially a popular district and county in Huizhou.

Investigating the indicator data for the three years from 2019–2021 and making a longitudinal comparison, it is found that the tourism development of the Huizhou region

in the three years shows an unstable state. It might be impacted by the recent pandemic of crowns. Overall, there has been a significant loss from 2019 to 2020, and even though the weather warms up in 2020 and 2021, there has been a 95% decrease in foreign visitors, which has had a negative effect on the profitability of the tourism sector. The Huizhou region, which has long been a well-liked destination for rural tourism, should take advantage of the tourism industry's significant recovery to raise the quality of its services and draw in a lot of domestic and foreign visitors.

This study of the variables influencing the quality of Huizhou's cultural and tourism services can help better identify the priority aspects and main breakthrough points in order for the high-quality development of Huizhou's tourism. Huizhou has a distinctive regional culture, mellow folk customs, and deep cultural connotations. Huizhou has an advantage in tourism resources, with hundreds of ancient villages, a large number of historical buildings, and excellent intangible cultural heritage, forming a unique "Huizhou culture". The local government should continue to protect and develop local culture, rationally plan inter-regional tourism resources, coordinate long-term cooperation of inter-regional rural resources, and create an excellent rural cultural tourism brand image in Huizhou. Establish the strategic direction of regional tourism in order to create a destination with a distinct culture and well-coordinated growth and to further the fusion of culture and tourism. Based on such resource advantages, the dynamic balance between tourism resources and the primary goal of efforts to raise the standard of culture and tourism services in the Huizhou region should be the environment. Secondly, the cooperation between the government and social organizations should be strengthened. To suit the needs of various tourists, public services should be made better, infrastructure should be built stronger, and there should be more hotels, particularly high-end hotels.

**Author Contributions:** Conceptualization, X.W. and Z.Y.; methodology, X.W. and Z.Y.; software, Z.Y.; validation, X.W., Z.Y. and Y.G.; formal analysis, Y.G.; investigation, X.W. and Z.Y.; resources, Z.Y.; data curation, Z.Y.; writing—original draft preparation, Z.Y.; writing—review and editing, Z.Y. and Y.G.; visualization, X.W. and Z.Y.; supervision, X.W. and Y.G.; project administration, X.W. and Y.G.; funding acquisition, X.W. All authors have read and agreed to the published version of the manuscript.

**Funding:** This research was supported by the 2022 Director's Fund Project of Anhui Institute of Culture and Tourism Innovation and Development (Project No. ACTZ2022ZD01), the "Digital Intelligence Rural Culture and Tourism Research and Innovation Team" (Project No. 2022AH010022), the China Postdoctoral Science Fund Project (Project No. 2023M730017), and the Youth Project of Anhui Academy of Social Sciences (Project No. QK202301).

**Institutional Review Board Statement:** Not applicable

**Informed Consent Statement:** All participants in this study provided their informed consent.

**Data Availability Statement:** The raw data supporting the conclusions of this article will be made available by the authors on request.

**Conflicts of Interest:** The authors have no conflicts of interest to declare.

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
