# Peer review of "Research on the Influencing Factors of Cultural and Tourism Service Quality in Huizhou Area"

_sustainability, doi:10.3390/su16135535_

Round 1

Reviewer 1 Report

Comments and Suggestions for Authors

Thank you for giving me this opportunity to review the manuscript entitled, “Research on the influencing factors of cultural and tourism service quality in Huizhou Area.”

Overall, the manuscript is well-written and present important results. I found some contents that are not well organized and some mistakes. I have some comments.  

1. Abstract

Abstract is well written.

2. Introduction

The introduction should clearly highlight the purpose of this study and contributions in the final paragraph.

3. Introduction

Information in introduction is valuable for understanding the manuscript overall. However, the contents of the introduction is not well organized. The content of the introduction is difficult to distinguish, and despite having good content, may be because there is no literature review. Different topics in introduction are combined into a single paragraph.

4. Research site

2.3. study area

There is not single citation regarding the study area. If the author use other materials for describing the research sites. There must be information sources.

5. Table1

It is very helpful for presenting indicators. If can, please add coding information and the data information. (data collection from 2019 to 2021)

6. title

4. Statistics and analysis

The title can be “results” in an academic paper.

7. Subtitle

Please revise the subtitle below

“44.2 Horizontal Comparison of Influencing Factors of Cultural and Tourism Service Quality in 383 Huizhou Region”

8. Table 3

What are the H1-8 in Table 3?

Since there is no literature review, the authors did not present their hypotheses or research questions more in detail.

9. Figure 3

Figure 3 images are too small to read for readers.

10. title

Please remove the “3.1.” before “5. Discussion”.

11. Table 5

Information described in Table 5 should be checked whether it is matched for readers.

Please add information if necessary.

12. Discussion

In the discussion, the results should be described as the authors explain them, without repetition, specifically for public, cultural services, and tourism industry- related factors. Please highlight the theoretical and practical implications in the discussion.

Reviewer 2 Report

Comments and Suggestions for Authors

The article is well-elaborated, and the topic is original. I would like to congratulate the authors for their work. There is only one comment that I would like to raise at the level of the methodology:

-          Clarification of Indicators: In the study, the authors mention the 20 evaluation indicators to assess service quality. I would suggest a more detailed explanation of how these indicators were selected, and also their relevance to the study would be very appreciated.

Reviewer 3 Report

Comments and Suggestions for Authors

I find the article titled "Research on the Influencing Factors of Cultural and Tourism Service Quality in Huizhou Area" interesting. I consider that the research questions are adequately addressed and resolved.

Next I will make some comments on the text.

1-The authors should provide greater justification of the importance of the study

2-It is necessary to justify the use of the applied methodology: why is it used? What advantages does it present over other methodologies with similar purposes?

3-In the text it says on one page that the period considered is 2019-2021 and on another page, 2019-2022. Which one is correct? Why is that period considered?

4- Section 2 does not seem to me to be correctly structured. It does not seem correct to me that the methodology and the area of ​​study are combined in the same section. It should be separated into two sections. On the one hand the methodology and on the other the area of ​​study.

5- Section 3 includes the indicators used in the analysis. I consider that in addition to listing them, their use must be justified.

6- I think that reference 14 is not correctly included in the bibliography. The reference in the text does not correspond to the reference in the bibliography section

7-What does equation 3 mean? How can it be interpreted?

8- The title of section 4 does not seem appropriate to me, it should be changed

Reviewer 4 Report

Comments and Suggestions for Authors

In their paper, the authors have focused on the service quality of cultural and tourism services in Huizhou region by using purely quantitative measures - available statistics.  They have developed an evaluation index system for the influencing variables of culture and tourism service quality by choosing 20 indicators from public service quality, cultural service quality, and tourism industry performance. I actually doubt that the quality can be measured by simply taking some available statistics on how many facilities, sites, hotels, services, etc. are. These figures provide very general picture about availability of service and tourism infrastructure but says absolutely nothing about the quality. I strongly believe that for the quality measurements some qualitative data are needed (e.g. tourist assessments, service providers self-assessments, etc.). I think this somehow should be integrated in the model/indicator system developed by the authors. Some research ethical aspects should be provided in the methodology description.

The authors refer to international studies and literature, the quality and amount of the sources is acceptable and appropriate for the scientific article. Discussion part and conclusions reflect the main results of the research.

Round 2

Reviewer 1 Report

Comments and Suggestions for Authors

Thank you for your specific explanations and revisions.

I would like to recommend the manuscript for publication.